# A Question Answering System for retrieving German COVID-19 data driven and quality-controlled by Semantic Technology

Andreas Both[1], Aleksandr Perevalov[1], Johannes Richard Bartsch[1], Paul Heinze[1], Rostislav Iudin[1], Johannes Rudolf Herkner[1], Tim Schrader[1], Jonas Wunsch[1], Ann Kristin Falkenhain[2], and René Gürth[3]

[1] Anhalt University of Applied Sciences, Köthen (Anhalt), Germany
[2] Federal Ministry of the Interior, Building and Community, Germany
[3] Informationstechnikzentrum Bund (ITZBund), Germany

**Abstract.** The COVID-19 pandemic is a showcase for a data-driven society. However, making the corresponding data available is not easy due to local characteristics and time-depending metrics. We present the Coronabot facilitating the access to German COVID-19 data capable of answering German and English questions. The component-based system is capable of understanding questions relating time & (even small) places in Germany. It is driven by RDF as all internal component interact with each other using RDF. Therefore, we are enabled to microbenchmark the component using SPARQL and therefore prove the quality requirements.

**Keywords:** Question Answering · User Interaction · COVID-19.

## 1 Introduction

During the Coronavirus pandemic numbers are presented to people on daily bases and are not easily comparable due to a strong dependency on the weekday. News portals provide them using text and maps for highlighting areas of spreading infections. However, local areas are of interest for citizens as restrictions (like curfews) depend on the metrics of the German districts. Despite other aspects this might lead to a misunderstanding and misinterpretation of the current situation by politicians, decision makers, and citizens. Additionally, debunking wide-spreading misinformation sometimes consumes precious resources if supporting data is needed. Hence, among other major challenges for societies, the pandemic is also a reference example of the need for good data accessibility providing the potential to increase transparency and trust.

Question Answering (QA) already proved its ability to increase the accessibility of data (typically encapsulated in a dialogue system, i.e., a chatbot). Using natural-language interfaces to collect information seems to be a reasonable option to offer data access. However, due to the characteristics of the pandemic not much data is available that can be used to train an end-to-end QA system. Additionally, in Germany the knowledge domain is connected to the German

districts which is an uncommon abstraction level for regular users as they typically expect an answer for their hometown (typically several cities are grouped together as districts). Finally, the quality of such a QA system needs to be very high due to the official characteristics of the offered data.

To address these issues we followed the Qanary methodology for our implementation [1]. The corresponding Qanary framework provides a component-oriented approach to rapidly integrate existing NLP/NLU functionalities. It uses RDF as internal knowledge representation. The Qanary components interact via SPARQL with the system-internal knowledge base (Qanary KB), s.t., traceability is leading to a constant quality control while using SPARQL to check whether the users' requests were correctly understood or not. In this paper we describe our demonstrator that was developed in collaboration with the Federal Ministry of the Interior, Building and Community of Germany and the Informationstechnikzentrum Bund (short: ITZBund) which is the government-owned software development unit for the German authorities. The demonstrator uses internally an RDF-driven approach. It encapsulates the functionality that will be integrated in the official COVID-19 chatbot of the German government[4].

## 2   Related Work

The architecture of data-driven dialogue systems is extended with on or more QA components [3]. Typically, QA systems are separated into 2 main paradigms: (1) Knowledge Base QA (KBQA) and (2) Open Domain QA (OpenQA). The KBQA systems are designed to give precise answers to natural language questions over structured data [2] while transforming questions into a corresponding query to a KB and then execute it in order to get an answer.

End-to-end QA also provides precise answers to natural language questions [5]. Such systems require large set of training data as they are based on large neural models and consequently are data-hungry. One of the major problems of all QA systems is that its components are being created from scratch all the time when a new system is developed [4]. To overcome the described problem, Qanary framework was created [1]. The core concept of the framework is not to invest double effort into creation of QA from scratch by reusing existing QA components.

## 3   Approach and Implementation

The intended QA system is demanded to provide support on place-based questions (e.g., "How many cases were reported in August in Dessau?"), where even very small places can be mentioned. Additionally, a time component is required leading to the possibility of fetching data for a specific time span (e.g., "How many people died in *February*?"). Moreover, the system has to answer questions that combine place and time dimensions. It is worth to mention, that the system has to work in two languages: German and English.

---

[4] https://chatbot.it.bund.de/

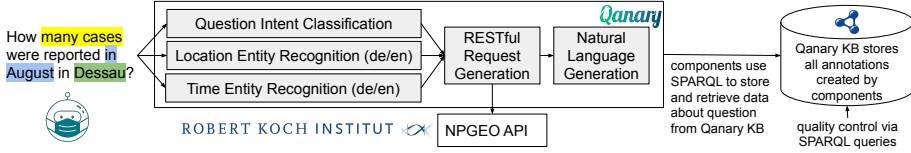

Fig. 1: System architecture overview (the actual system contains 11 components)

Following the Qanary methodology, typical tasks of the question processing are identified to model the components of the QA system. Each component will execute its dedicated task, i.e., creating new information about the given question. Finally, each component is storing the computed information in the Qanary triplestore which is representing the global memory of the information computed while analyzing a question. In the Qanary methodology this RDF data is called an *annotation of the given question.*

As the data is stored as RDF the annotations are modeled using the Qanary vocabulary which is a lightweight extension of the Web Annotation Data Model (WADM)[5]. While retrieving the stored annotations the following components can retrieve any information that was previously computed while analyzing the given question. Here, an API[6] needs to be requested via a RESTful request to a Web service capable of providing the official data since the pandemic started.

According to this approach the following Qanary components were designed and implemented (see Figure 1): *Language Detection* as the component detecting whether a German or English question was asked, *Question Intent Classification* recognizes one of the following 6 classes: location, time, location and time (for infection or death cases), *Location Entity Recognition* detects location entities and links the detected text span to entity ID[7], *Time Entity Recognition* recognizes time entities in the question and converts it to a normalized date representation, *RESTful Request Generation* generates and executes an NPGEO API request, *Natural Language Generation* provides a textual answer. The components are controlled by the out-of-the-box Qanary pipeline[8] (the reference implementation of the Qanary methodology). Hence, while a question is processed a RDF graph is created reflecting all information that was computed by the QA process. To fulfill the quality demands an extensive test set of 40.000 questions were defined that is evaluated while reusing the RDF data stored in the Qanary triplestore, i.e., each test question is executed and thereafter several SPARQL queries are used to check if the expected annotations were created in the virtual graph that is corresponding the current question (so-called *graph of RDF annotations*).

---

[5] W3C Recommendation, 2017-02-23 (cf. `http://www.w3.org/TR/annotation-model`)
[6] NPGEO API of Robert Koch institute, the German federal government agency responsible for disease control and prevention: `https://npgeo-corona-npgeo-de.hub.arcgis.com/datasets/dd4580c810204019a7b8eb3e0b329dd6_0`
[7] The following Germany-specific entities are detected: federal states (16), counties (412), cities + communities (11806) where the latter are retrieved from Wikidata.
[8] cf. `https://github.com/WDAqua/Qanary`

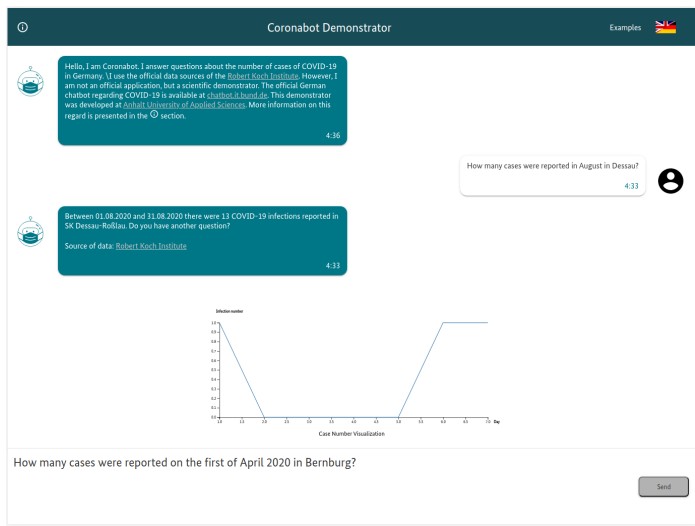

Fig. 2: Coronabot screenshot and RDF graph of annotations.

## 4    Conclusions

In this work, we showed a Linked Data driven and component oriented approach for creating dialogue systems. The domain of the system is consultation of German residents on Coronavirus pandemic statistics (Coronabot). Additionally, the Coronabot works with two languages: German and English. The system was developed according to Qanary methodology which demonstrated its flexibility for developing and evaluation processes. Since the QA system is RDF-based, it also offers the great advantage of processing traceability and thus end-to-end quality control. The demonstrator is available at `http://coronabot.ins.hs-anhalt.de/`.

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

# Description of the Demonstration

The demo is available at http://coronabot.ins.hs-anhalt.de/. Please see the menu item "examples" to get familiar with the support questions.

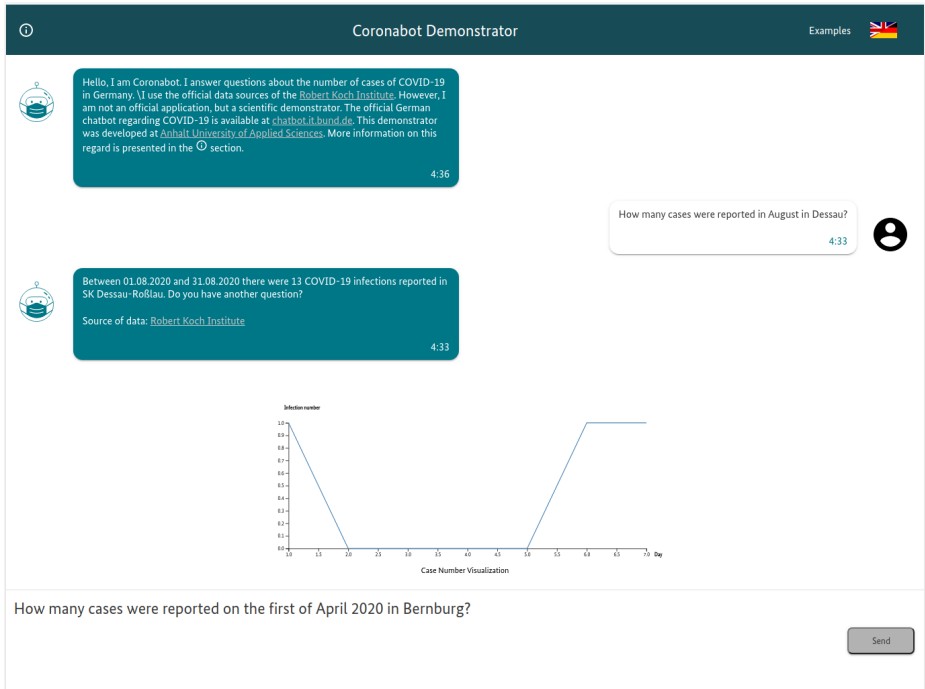

During the demo we will show the processing of questions and the quality control. The latter follows two aspects. (1.) The explorative approach where using RDF is the most important issue as it allows a later analysis of the whole Question Answering process for a particular question. (2.) The permanent quality control. For this purpose we will use the Stardog studio for interactively executing SPARQL queries while validating the created annotations during the QA process. Additionally we will provide a batch tool that allows the execution of predefined questions and executes SPARQL (ASK) queries for validating the expected behavior for

**Explorative Approach:** In the following a subset of the RDF graph is shown that was created during the processing of the question "How many cases were reported in August in Dessau?" (so-called *graph of annotations*). The interactive tool will also be available for public use at the time of the conference.

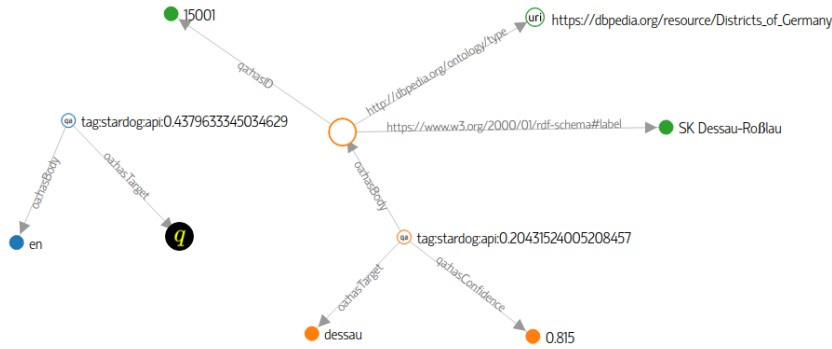

**Permanent quality control:** We will show a tool (will be open-source at the time of the conference) capable of analyzing a given list of questions with a given list of SPARQL queries. The SPARQL queries are used to trace based on the given question the processing while querying the Qanary KB (see Fig. 1). Hence, even while exchanging components the quality can be evaluated also pointing directly to the component that is not computing the expected quality for a particular question. In the demo we will show the overall quality of the pipeline and also identify precisely what questions have failed and why.

A subset of the test definitions (several of these test configuration exist in a JSON file format):

```
{
    "qanary": {
"system_url":
"http://webengineering.ins.hs-anhalt.de:44040/startquestionansweringw
ithtextquestion",
        "componentlist": [
            "LD-Shuyo",
            "coronabot-question-classification",
            "coronabot-named-entity-recognition-english-time-ml",
            "coronabot-named-entity-recognition-german-time-ml",
            "coronabot-named-entity-recognition-location-spacy-eng",
            "coronabot-query-generation",
            "coronabot-data-acquisition"
        ],
                                        "qanary_triplestore_endpoint":
"http://webengineering.ins.hs-anhalt.de:44041",
        "qanary_triplestore_database": "...",
        "qanary_triplestore_username": "...",
        "qanary_triplestore_password": "..."
    },
    "validation-sparql-templates": [
        "is-location-id-correct.sparql",
        "is-location-type-correct.sparql",
```

```
        "was-language-identified.sparql",
        "is-language-correct.sparql"
    ],
    "custom-validation": "query-rki-api",
    "tests": [
        {
                "question": "Wie viele Fälle gibt es in Stdtkreis
Delmenhorst?",
            "replacements": {
                "QUESTION_CLASS": "infection_location",
                "LANGUAGE": "de",
                "NUMBER_TYPE": "qa:AnnotationOfInfectionNumber",
                "LOCATION_ID": "03401",
                "LOCATION_TYPE": "Districts_of_Germany",
                    "ANSWER_TEXT": "There are currently \\\\d+ cases
reported for \\\\w+."
            }
        },
        {
            "question": "Wie viele Fälle von Covid 19 hat Cham?",
            "replacements": {
                "QUESTION_CLASS": "infection_location",
                "LANGUAGE": "de",
                "NUMBER_TYPE": "qa:AnnotationOfInfectionNumber",
                "LOCATION_ID": "09372",
                "LOCATION_TYPE": "Districts_of_Germany",
                    "ANSWER_TEXT": "There are currently \\\\d+ cases
reported for \\\\w+."
            }
        },
        {
            "question": "Infektionen gosenheim",
            "replacements": {
                "QUESTION_CLASS": "infection_location",
                "LANGUAGE": "de",
                "NUMBER_TYPE": "qa:AnnotationOfInfectionNumber",
                "LOCATION_ID": "09163",
                "LOCATION_TYPE": "Districts_of_Germany",
                    "ANSWER_TEXT": "There are currently \\\\d+ cases
reported for \\\\w+."
            }
        },
        {
                "question": "Wie viele Fälle von Covid 19 hat Rhein
Neckar Kreis?",
            "replacements": {
                "QUESTION_CLASS": "infection_location",
                "LANGUAGE": "de",
                "NUMBER_TYPE": "qa:AnnotationOfInfectionNumber",
                "LOCATION_ID": "08226",
                "LOCATION_TYPE": "Districts_of_Germany",
```

```
                    "ANSWER_TEXT": "There are currently \\\\d+ cases
reported for \\\\w+."
            }
        },
        {
            "question": "Fälle in chwerin?",
            "replacements": {
                "QUESTION_CLASS": "infection_location",
                "LANGUAGE": "de",
                "NUMBER_TYPE": "qa:AnnotationOfInfectionNumber",
                "LOCATION_ID": "13004",
                "LOCATION_TYPE": "Districts_of_Germany",
                    "ANSWER_TEXT": "There are currently \\\\d+ cases
reported for \\\\w+."
            }
        },
        {
            "question": "Fälle in LK Hochtaunuskreis?",
            "replacements": {
                "QUESTION_CLASS": "infection_location",
                "LANGUAGE": "de",
                "NUMBER_TYPE": "qa:AnnotationOfInfectionNumber",
                "LOCATION_ID": "06434",
                "LOCATION_TYPE": "Districts_of_Germany",
                    "ANSWER_TEXT": "There are currently \\\\d+ cases
reported for \\\\w+."
            }
        },
        {
            "question": "Wie viele Fälle gibt es in Heilbronn?",
            "replacements": {
                "QUESTION_CLASS": "infection_location",
                "LANGUAGE": "de",
                "NUMBER_TYPE": "qa:AnnotationOfInfectionNumber",
                "LOCATION_ID": "08121",
                "LOCATION_TYPE": "Districts_of_Germany",
                    "ANSWER_TEXT": "There are currently \\\\d+ cases
reported for \\\\w+."
            }
        }, ...
```

In the following you see the SPARQL queries used in the test specification:

is-location-id-correct.sparql

```
# verify correctness of annotated location
PREFIX oa: <http://www.w3.org/ns/openannotation/core/>
PREFIX rdf: <http://www.w3.org/1999/02/22-rdf-syntax-ns#>
PREFIX qa: <http://www.wdaqua.eu/qa#>
PREFIX dbr: <http://dbpedia.org/resource/>

ASK
FROM <GRAPHID>
WHERE {
    ?annotationId rdf:type qa:AnnotationOfInstanceLocation .
    ?annotationId oa:hasBody ?location_id .
}
HAVING(STR(?location_id) = "LOCATION_ID")
```

is-location-type-correct.sparql

```
# verify correctness of annotated location
PREFIX oa: <http://www.w3.org/ns/openannotation/core/>
PREFIX rdf: <http://www.w3.org/1999/02/22-rdf-syntax-ns#>
PREFIX qa: <http://www.wdaqua.eu/qa#>
PREFIX dbr: <http://dbpedia.org/resource/>
PREFIX dbo: <http://dbpedia.org/ontology/>

ASK
FROM <GRAPHID>
WHERE {
    ?annotationId rdf:type qa:AnnotationOfInstanceLocation .
    ?annotationId oa:hasBody [
        dbo:type ?location_type
    ] .
}
HAVING(STR(?location_type) = "dbr:LOCATION_TYPE")
```

was-language-identified.sparql

```
# language recognition component needs to identify a language
PREFIX oa: <http://www.w3.org/ns/openannotation/core/>
```

```
PREFIX rdf: <http://www.w3.org/1999/02/22-rdf-syntax-ns#>
PREFIX qa: <http://www.wdaqua.eu/qa#>

ASK
FROM <GRAPHID>
WHERE {
    ?annotationId rdf:type qa:AnnotationOfQuestionLanguage .
    ?annotationId oa:hasBody ?body .
    ?annotationId oa:hasTarget ?target .
    ?annotationId oa:annotatedBy ?component.
}
HAVING (COUNT(distinct ?annotationId) > 0)
```

### is-language-correct.sparql

```
# verify correctness of annotated location
PREFIX oa: <http://www.w3.org/ns/openannotation/core/>
PREFIX rdf: <http://www.w3.org/1999/02/22-rdf-syntax-ns#>
PREFIX qa: <http://www.wdaqua.eu/qa#>
PREFIX dbr: <http://dbpedia.org/resource/>

ASK
FROM <GRAPHID>
WHERE {
    ?annotationId rdf:type qa:AnnotationOfQuestionLanguage .
    ?annotationId oa:hasBody ?language .
}
HAVING(STR(?language) = "LANGUAGE")
```