# OpenReview forum: "A Question Answering System for retrieving German COVID-19 data driven and quality-controlled by Semantic Technology"
_eswc-conferences.org/ESWC/2021/Conference/Poster_and_Demo_Track — Submitted to ESWC2021 P&D_

### Official Review · AnonReviewer1 · 2021-04-11
**Review of Paper40**

**Rating:** 6
**Confidence:** 4

**Review:**

Quality: fair

Clarity: good

Originality: fair

Significance of this work: good

Pros:
1.	This paper presents a Coronabot that can facilitate answering German and English questions of COVID-19 by Qanary methodology.
2.	The system architecture is available and the interaction demonstrator can be assessed.
3.	This Coronabot is friendly for users (e.g., some trend charts) and the German COVID-19 data is Comprehensive.

Cons:
1.	The ability for answering the open question is not well. For example. When I ask one question “what is COVID-19?”. The Coronabot replies: “The location you asked for could not be identified. Which what is do you mean?”
2.	Lack some quantitative assessments for this work.

**Anonymity:**

Yes, I would like my review to remain anonymous.

---

### Official Review · AnonReviewer3 · 2021-04-13
**hard to understand; require validation**

**Rating:** 3
**Confidence:** 4

**Review:**

This paper proposes Coronabot, a chatbot for retrieving German COVID-19 data. The core of this proposal is based on the Qanary methodology which:
 1. Recognize the language of the input.
 2. Recognize specific type of entities (e.g., locations and time expressions).
 3. Link them to a Knoledge Graph (KG).
 4. Obtain the answer using an external API (NPGEO).
 5. Generate the natural language answer with the information found.

I consider this paper as a work in progress. This paper is hard to understand; so far, I'm not sure what its contribution is. The authors refer in different parts of the papers to different ideas: "we present the Coronabot", "in this paper we describe our demonstrator". Besides, in conclusion, the authors says:

 "The domain of the system is consultation of German residents on Coronavirus pandemic statistics (Coronabot)"

which makes me think Coronabot is the domain of the system proposed. In my view, the Qanary methodology has a significant part of the process, and this paper only proposes a kind (no clear) validation of the Qanary's output. I also recommend clearing the Qanary process itself (in case you keep it in the paper). Quoting the paper,

- Language Detection: "as the component detecting whether a German or English question was asked"
- Question Intent Classification: "recognizes one of the following 6 classes: location, time, location and time (for infection or death cases)"
- Location Entity Recognition: "detects location entities and links the detected text span to entity ID"
- Time Entity Recognition: "recognizes time entities in the question and converts it to a normalized date representation"
- RESTful Request Generation: "generates and executes an NPGEO API request"
- Natural Language Generation: "provides a textual answer"

The "Question Intent Classification" should recognize six classes, but only two were specified (location and time). There is an overlapping in which entities the components recognize that could be intentional, but there is no explanation about that, i.e., "Question Intent Classification" and "Location Entity Recognition" recognize entities of locations simultaneously. "Question Intent Classification" and "Time Entity Recognition" recognize time expression entities simultaneously. In the "RESTful Request Generation" description, I recommend including some references for the NPGEO API and give some details about it. I also recommend extending the definition of the "Natural Language Generation" component.

Some other comments next:
 - typo "location, time, location and time"
 - Figure 1 shows the system architecture, but the authors remark that "the actual system contains 11 components". It was not clear to me if components of the Qanary or your proposal were missing. In case they have referred to components of your proposal, I would recommend including them.
 - Although this is a demo paper, I would expect to see an evaluation of the system to demonstrate its performance.

**Anonymity:**

Yes, I would like my review to remain anonymous.

---

### Official Review · AnonReviewer4 · 2021-04-15
**Very interesting work which unfortunately is not presented very well: unclear problem statement and unjustified choices**

**Rating:** 4
**Confidence:** 3

**Review:**


This paper presents an RDF-based chatbot application to retrieve regionally specific information about COVID-19 measures.

The paper is above page limit, however, the 4-page content is self-standing and the appendix could easily be moved to a provided website to fix this issue. I consider the appendix to be the website, therefore my review is based on the 4 page content and would remain the same if the appendix was removed.

## Summary

The context of the paper is very relevant and good explained.
However, the problem statement and need for the solution is not well explained, using RDF and SPARQL as solution seems to come out of the blue.
Additionally, there seems to be an existing chatbot, why is the current work performed? What are issues with the existing chatbot,
or which issues cannot be tackled with the existing chatbot such that the presented RDF solution is needed to extend it?

Although very relevant and actually interesting work,
the paper suffers from severe presentation issues.

The performed work is very interesting and I believe the solution is capable of good results,
unfortunately from the paper all this is unclear.
I encourage the authors to rework the paper -- currently more a system description -- to have a clear problem statement and provide justified design decisions.

Please find my detailed review below.

## Introduction

I'm not fully convinced with the reasoning provided for QA systems:

> "Question Answering (QA) already provided its ability to increase the accessibility of data ..."

There is no reference, thus I am forced to believe the authors. Ideally a citation and also refining the statement to *how exactly* it increases *what* part of accessibility

> "Using natural-language interfaces to collect information seems to be a reasonable option to offer data access"

Again no reference and not specific enough. *Why* is it a reasonable option? Concrete queries are more precise and quicker to process.
 Is it reasonable because it is easier to use by ordinary-citizens? If so this should be mentioned, but then ideally with a reference to a user study too.
 NLP also introduce lots of problems, most chatbots I have used so far where not that useful.

> "However, due to the characteristics of the pandemic not much data is available that can be used to train an end-to-end QA system"

Why not? And which characteristics? To me it seems there are lots of data, as mentioned by the authors there are news portals and "numbers are presented to people on daily bases"
 This should be clarified.

> "To address these issues we followed the Qanary methodology fo rour implementation"

I miss again the *why*? Why the qanary methodology, what does it make so relevant for this use case? Is it commonly used? Was it effectively used already in the past for chatbots?


> "[The Qanary framework] uses RDF as internal knowledge representation"

Why is this positive? Do we need RDF and SPARQL? Couldn't the problem be solved with a relational database?
Ideally here would be some reasoning why the presented problem demands RDF as a solution.
(Later in section 3 I read that both temporal and spatial data are needed, this could be mentioned early on such that it becomes apparent to a reader why RDF is a good choice)

## Related work

Explanations are given and relevant related work is provided.
However, in my opinion a bit too late, a few provided arguments may have fit better to the introduction to justify choices such as why Qanary is used.

## Approach and Implementation


> "each component is storing the computed information in the Qanary triplestore"

What are the components? How do they produce RDF and how will it be stored in the triple store?
Some more information are needed to accurately describe the approach

> "Here, an API needs to be requested"

I think the authors mean that data *from an* API needs to be requested. Otherwise it could be misunderstood that an actual API is created on the fly based on some desccription.

> "where the latter are retrieved from Wikidata"

From where are the former ones retrieved from if not Wikidata?


**The figures**

Figure 1 is hard to understand. Temporal, spatial and a third component are marked in colors, yet these colors are not reused in the corresponding boxes.
It might be more clear to not use colors and possibly reformat a bit.

What is retrieved by the NPGEO API, and does the NPGEO API comes from the Robert Koch Institut which is mentioned next to it? (According to a footnote it is, but it is not very clear from the figure)

The caption of figure 1 is not very descriptive ("system architecture overview") and even confusing "(the actual system contains 11 components"): what are the 11 components of the actual systems?
Why aren't those 11 components showed in the overview?

Figure 2 can barely be read.
The caption of figure 2 is not very descriptive either. Overall this figure does not contribute much information while taking space.
What does the graph in the second figure show? What are the questions and what the answers?

## Conclusion

In the introduction it was mentioned that there is already an official COVID-19 chatbot of the German government,
and that presented functionality will be integrated there.

But why?
If there exists already a chatbot, why is the presented extension needed?
What does it contribute? Where are existing problems? How well addresses the proposed work potential existing issues?

## Minor

* In the abstract: "all internal component" -> components
* A footnote for "districts" could provide the actual German name to avoid ambiguity.

**Anonymity:**

Yes, I would like my review to remain anonymous.

---

### Decision · Program_Chairs · 2021-04-19

Reject